

# Preflight Calibration of the Chinese Environmental Trace Gases Monitoring Instrument (EMI)

## MinJie Zhao, FuQi Si, HaiJin Zhou, ShiMei Wang, Yu Jiang, WenQing Liu

Key laboratory of Environmental Optics and Technology, Anhui Institute of Optics and Fine Mechanics, Chinese
Academy of Sciences, Hefei, 230031, China

*Correspondence to:* FuQi Si (sifuqi@aiofm.ac.cn)

## Abstract

The Environmental trace gases Monitoring Instrument (EMI) is a nadir-viewing wide-field imaging spectrometer, aiming to quantify the global distribution of tropospheric and stratospheric trace gases, which is planned to be launched in 2018. The selected wavelength bands for EMI are UV1(240-315nm), UV2(311-403nm), VIS1(401-550nm) and VIS2(545-710nm), the spectral resolution is 0.3-0.5nm, and the swath is about 114 degrees wide to achieve one-day global coverage. The preflight calibration is discussed in this paper. Tunable laser and rotating platform are adopted for the EMI wavelength calibration of the whole field of view. The accuracy of the wavelength calibration is better than 0.05nm. In addition, the calibration data in the Sun calibration mode shows that the same calibration results are obtained compared with the Earth observation mode. In order to investigate the influence of in-orbit thermal-vacuum conditions on the EMI, the thermal vacuum test is performed, and the EMI spectral response changes with pressure, optical bench temperature and CCD temperature are obtained. For the radiometric calibration of UV1, the diffuse plate with a 1000W xenon lamp is chosen, which produces sufficient ultraviolet output. And the integrating sphere system with tungsten halogen lamp is selected for the UV2, VIS1 and VIS2. The accuracy of the radiance calibration is 4.53%(UV1), 4.52%(UV2), 4.31%(VIS1) and 4.30%(VIS2). The goniometry correction factor and irradiance response coefficient of the EMI are also calibrated on the ground for the in-orbit calibration of the solar. As the effect of Signal to Noise ratio(SNR) on the retrieved results, a SNR model of the EMI is introduced, and the EMI in-orbit SNR is estimated using the SNR model and the MODTRAN radiance model.

## 1 Introduction

A series of space-borne spectrometers like GOME[*A.Hahne et al., 1993*], SCIAMACHY[*S. Noel et al., 1998*], GOME-2[*Rosemary Munro, et al., 2016*] and OMI[*Pawan K Bhartia et al., 2006*] have been successfully applied to the global monitoring of atmospheric trace gas distributions. These instruments measure sun radiance backscattered from the Earth atmosphere in the UV-VIS wavelength range. The TROPOMI builds upon the heritages of the SCIAMACHY and the OMI instruments, which was launched in 2017



on ESA's Sentinel 5 precursor satellite[*Rovert Voors et al., 2012*].

The Environmental trace gases Monitoring Instrument (EMI) is a space-borne nadir-viewing
wide-field imaging spectrometer, which is used to obtain global distributions of tropospheric and
stratospheric trace gases(e.g. NO2, O3, HCHO, SO2) at high spatial and spectral resolution. The EMI is
planned to be launched in 2018.

The EMI has four spectral channels(UV1,UV2,VIS1,VIS2) ranging from 240 nm to 710 nm. Each
channel adopts Offner imaging spectrometer, and two-dimensional charge-coupled device detectors.
The EMI enables an instantaneous field of view (FOV) of 114° (corresponding to a 2600 km broad
swath on the Earth's surface), the space resolution is either 8km/12km(UV/VIS channel) or
48km(UV, VIS channel) at nadir, depending on the electronic binning factor, see table 1. And one-day
global coverage can be realized. The anticipated lifetime of EMI is eight years, and its properties are
shown in Table 1.

Table 1. EMI instrument properties

| | |
|---|---|
| **Spectral range** | UV1: 240-315nm;   UV2: 311-403nm |
| | VIS1: 401-550nm;   VIS2: 545-710nm |
| **Spectral sampling** | UV1: 0.08nm;   UV2: 0.09nm |
| | VIS1: 0.12nm;   VIS2: 0.13nm |
| **Spectral resolution(FWHM)** | 0.3-0.5nm |
| **Telescope swath IFOV** | 114 degrees(2600 km on the ground) |
| **Telescope flight IFOV** | 0.5 degrees(6.5 km on the ground) |
| **CCD detectors** | UV: $1072 \times 1032$ (spectral $\times$ spatial) pixels |
| | VIS: $1286 \times 576$ (spectral $\times$ spatial) pixels |
| **Ground pixel size at nadir** | 13km $\times$ 48km(electronic binning factor |
| | UV:24,VIS:16) |
| | 13km $\times$ 8km(UV, binning factor 4) |
| | 13km $\times$ 12km(VIS, binning factor 4) |
| **Orbit** | Polar, sun-synchronous; Orbit period: 98 minutes 53 seconds, Ascending node equator crossing time: 13:30 PM |

The optical layout of the EMI is shown in Fig.1. The EMI consists of a telescope and four



spectrometers.

    The telescope provides an instantaneous field of view of 114° in the swath direction and of 0.5° in the flight direction, which yield an overall ground coverage of about 2600km by 6.5km at an altitude of 50  705km. The spatial resolution in the swath direction depends on the electronic binning factor, in the flight direction depending on the CCD integral time. Four Offner imaging spectrometers are adopted by EMI, each spectrometer with a convex grating and a 2-dimensional CCD detectors. The Offner imaging spectrometer is easy to be miniaturized and lightweight, and is suitable for the development of space technology. It is also suitable for high spatial and spectral resolution detection systems. The EMI cover 55  240-710nm range with the spectral resolution 0.3-0.5nm.

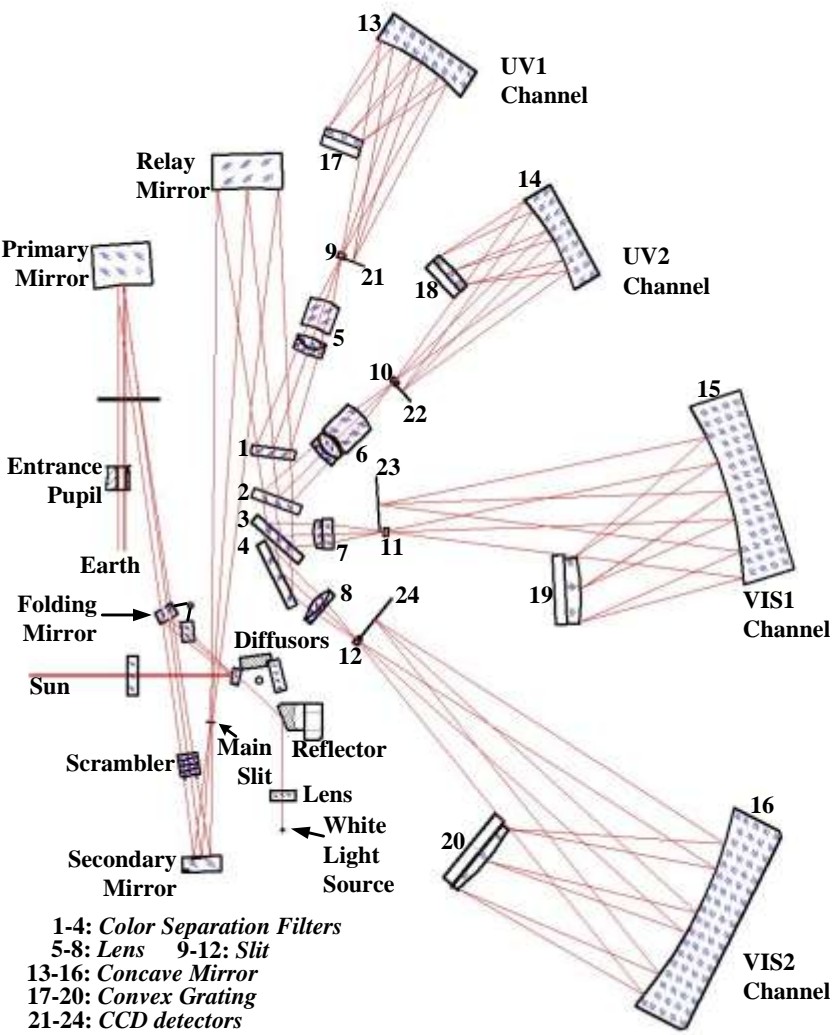

1-4: *Color Separation Filters*
5-8: *Lens*    9-12: *Slit*
13-16: *Concave Mirror*
17-20: *Convex Grating*
21-24: *CCD detectors*

Fig.1. Optical layout of the EMI instrument



One observation mode and two calibration modes are include. The observation mode is used to detect the atmospheric scattering light, the two calibration modes are for in-orbit calibration.

In the observation mode, the Earth radiance enters the telescope via the entrance pupil, and is imaged on the main slit after reflection by the primary and secondary mirror. A polarization scrambler is located before the secondary mirror, which is used to make the EMI insensitive to the polarization state of the incident light. Behind the main slit a relay mirror reflects the incident light on the color separation filter 1-4. The color separation filter 1 reflects the 240-315nm range of the spectrum to the UV1 channel

and transmits the rest of spectrum to the color separation filter 2. As a result, 311-403nm, 401-550nm, and 545-710nm range of the spectrum are reflected to the UV2, VIS1 and VIS2 channel by the filter 2-4. The spectrum form the filters is imaged on the spectrometer slit 9-12(10mm×60μm) via lens 5-8. And then final dispersion is achieved by the convex grating 17-20 after reflection by the concave mirror 13-16, that is used in first order. Finally the spectrum is imaged onto 2- dimensional(spectral and spatial

dimension) CCD detectors 21-24.

   First of the calibration modes is the solar calibration, the sun spectrum observed by this mode is used to perform accurate wavelength calibrations and to normalize the Earth spectra in order to obtained the absolute Earth reflectance spectra. The solar radiation enters the instrument through a mesh(transmission 10%) by opening the solar aperture mechanism, and is diffused by the selected

diffuser. Light from the diffusers illuminates the folding mirror, and is reflected to the telescope optical path. The folding mirror in this position blocks the Earth radiance form primary mirror. The EMI equipped with one surface reflectance aluminum diffuser(40 mm×16 mm) and one quartz volume diffuser(QVD, 40 mm×16 mm×6 mm) , which consists of 6-mm thick quartz ground on both sides and coated with aluminum on the backside. Besides its use for radiometric calibration, the QVD is used

once per day to provide the solar reference spectrum, this is because considerably less structure are introduced by QVD than aluminum diffuser [*Ruud Dirksen et al., 2004, Johan de Vries et al., 2005*]. The aluminum diffuser is mainly used for monitoring of optical degradation behavior in space, which is performed monthly.

   The second calibration mode is the white light source calibration, a quartz tungsten halogen white

light source (WLS, 6 V, 10 W) is used to monitoring of the CCD detector properties. The light form WLS travel through the transmission diffuser and is reflected to the telescope optical path.

## 2 Preflight calibration

The EMI detection ability needs matching the changes of the Earth radiance, thus the instrument can obtain better data from in orbit. In order to get the response performance of the instrument,

high-precision spectral and radiometric calibration are required on the ground[*A. Perez Albinana et al., 2002, Marcel Dobber et al., 2006, B. Ording et al., 2016, Quintus Kleipool, et al., 2018*].

### 2.1 Spectral calibration

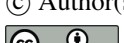


The spectral calibration is performed in the Earth observation mode(EOM) during laboratory calibration. The calibration results of the EOM can be applied to Solar calibration mode(SCM), see 2.3 section. The

employed tunable laser (OPOTEK: RADIANT) has a output spectrum range 193-410nm and 410-2500nm, which can cover 240-710nm of the EMI. with the wavelength precision 10pm. In addition, the spectral calibration is carried out in clean room, which can reduce the influence of temperature and humidity.

The spectral calibration is needed in spectral and spatial dimension. The tunable laser output

wavelength space is 5nm for UV2 channel, and is 10nm for UV1, VIS1 and VIS2 channel calibration in the spectral dimension. The spectral lines have full widths at half maximum that are typically an order of magnitude lower than the EMI spectral resolution, thus providing basically delta inputs to the EMI instrument in the wavelength dimension, as a result, the influence of the slit function of the laser is removed. In the spatial dimension, the instrument has to be rotated in 21 steps according to the 5.5°

interval to cover the full FOV. The spectral calibration and dark background data are recoded.

The wavelength calibration of the EMI instrument is given by

$$\lambda_{i,j} = \sum_{m=0}^{N} c_{k,j} \cdot p^k$$

where $\lambda$ is the wavelength of the pixel, $i$ is the column number, $j$ is the row number, and $c_{m,j}$ are the

wavelength calibration polynomial coefficients. $N$ is the order of the polynomial, which is 3 for the EMI

wavelength calibration. The spectral lines of laser distribute uniformly in the spectral dimension, which ensure the polynomial fitting precision. The four channel wavelength calibration of center field of view(CFOV) in spectral dimension are shown in Fig.2.

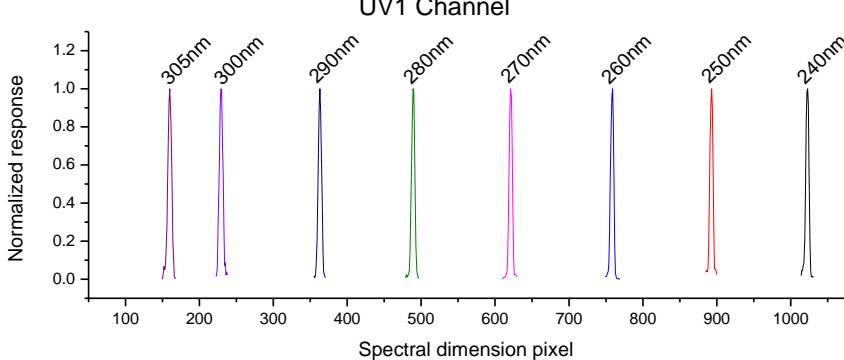

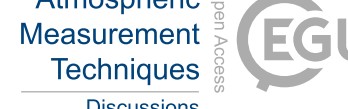



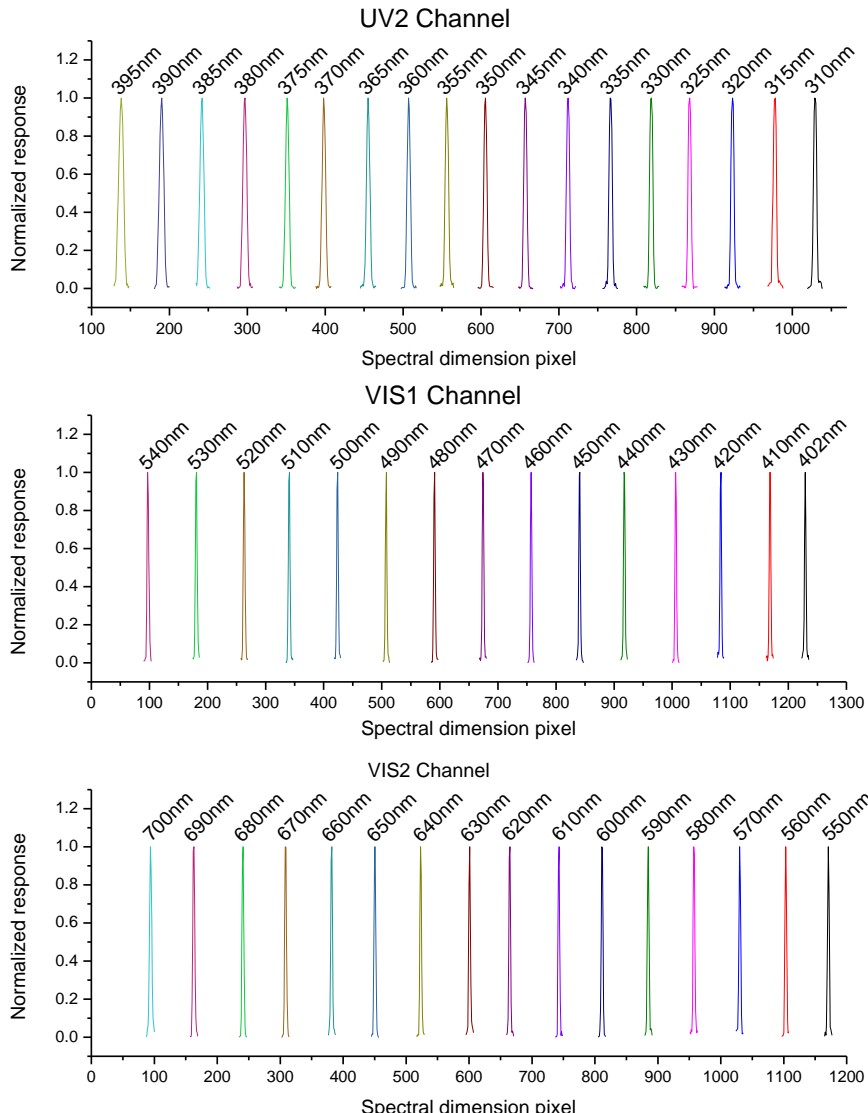

Fig.2. EMI center field of view wavelength calibration for each channel. The upper panel presents
the UV channel, the lower panel presents the VIS channel. The spectral responses are normalized.

The CFOV spectral range of each channel are shown in table 2, the spectral range in other field of
view are discussed latter.





Table 2. CFOV spectral range

| Channel | Spectral Range/nm |
|:---:|:---:|
| **UV1** | *236.44～317.28* |
| **UV2** | *306.08～407.12* |
| **VIS1** | *395.50～552.63* |
| **VIS2** | *534.63～712.90* |

The spectral calibration in spatial dimension are shown in Fig.3. It can be seen that the smile effect in spatial dimension exists in each channel, the wavelength position on the detector array varies with different field of view[*P. S. Barry et al., 2002, Robert A et al., 2003, Luis Guanter et al., 2006*]. The wavelength in marginal field of view shift to long wave for UV channel and shift to short wave for VIS channel.

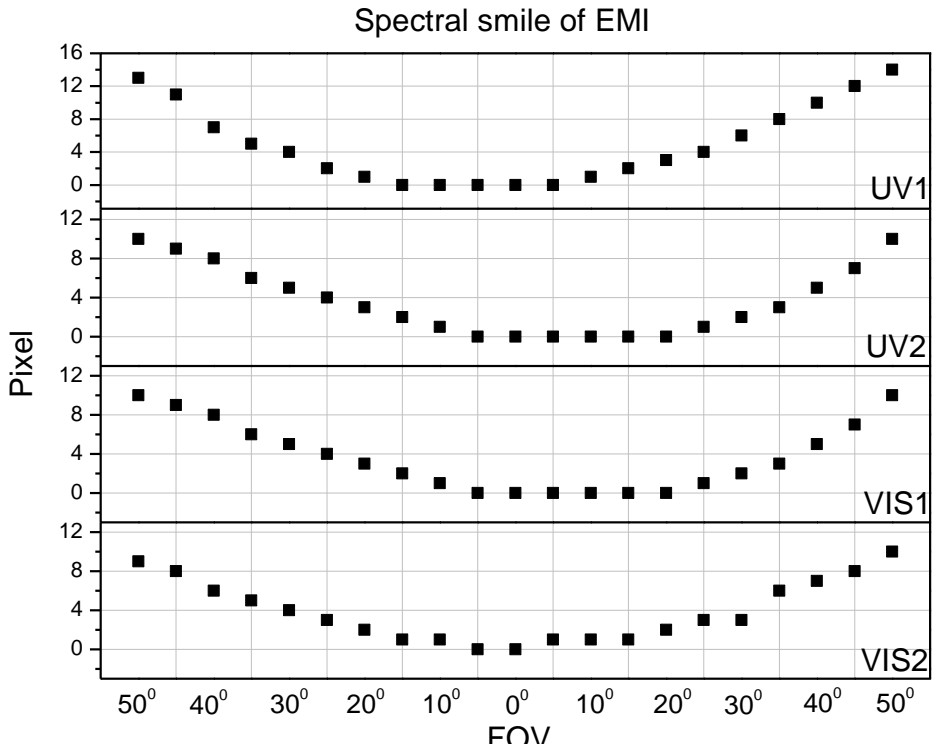

Fig.3. Spectral calibration in the spatial dimension.

The spectral response of EMI can be considered as the Gaussian function, its FWHM(full width at half maximum) is known as the spectral resolution of the spectrometer channels. The FWHM of the EMI by Gaussian fitting is shown in table 3.





Table 3. FWHM

| FOV | UV1/nm | UV2/nm | VIS1/nm | VIS2/nm |
|---|---|---|---|---|
| **50°** | 0.44 | 0.45 | 0.34 | 0.49 |
| **40°** | 0.39 | 0.39 | 0.29 | 0.39 |
| **30°** | 0.40 | 0.38 | 0.29 | 0.40 |
| **20°** | 0.42 | 0.43 | 0.31 | 0.39 |
| **10°** | 0.42 | 0.47 | 0.33 | 0.39 |
| **0°** | 0.43 | 0.49 | 0.34 | 0.40 |
| **10°** | 0.41 | 0.46 | 0.34 | 0.38 |
| **20°** | 0.38 | 0.41 | 0.32 | 0.34 |
| **30°** | 0.36 | 0.36 | 0.34 | 0.30 |
| **40°** | 0.38 | 0.36 | 0.38 | 0.28 |
| **50°** | 0.45 | 0.43 | 0.48 | 0.34 |

The overall accuracy of the spectral calibration is determined by three mayor factors, firstly by the accuracy of laser output wavelength, which is better than 0.01nm, secondly by the stability of the EMI spectral response, which is determined by 20 spectral response data from the same laser output line (<0.014nm),   thirdly by fitting method(using the least square method), the accuracy of the polynomial fitting is about 0.040nm and the Gaussian fitting is about 0.020nm. The final accuracy of the
wavelength calibration is better than 0.05nm, and the spectral response function is better than 0.03nm.

**2.2 Thermal vacuum test**

The spectral calibration discussed above is performed in atmospheric environment, which can provide detailed spectral response characteristics. In order to obtain the difference between atmospheric and vacuum environment and to obtain the spectral response characteristics in thermal vacuum 
conditions(EMI in-flight conditions), the thermal-vacuum test is performed, see Fig.4. The EMI instrument views the mercury argon lamp through the thermal-vacuum chamber window. Because the limit of a rotational device and the window size, the center field of view of EMI is measured in the thermal-vacuum chamber.





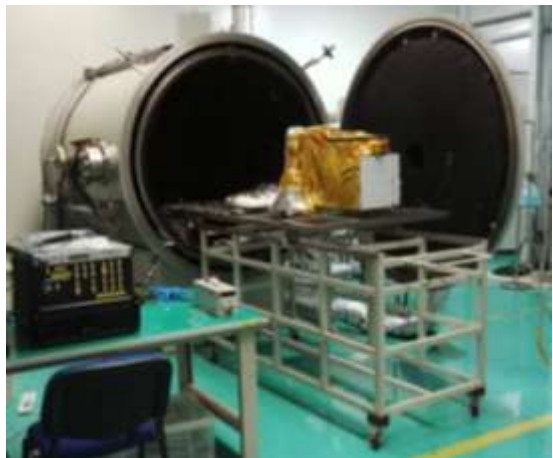

Fig.4. Thermal vacuum test of the EMI.

The thermal vacuum conditions include pressure, optical bench temperature and CCD temperature:

**Pressure:**

*AE*: Atmospheric Environment

*PV* : Pumping Vacuum

*NFP*: Nitrogen Filling Process

**Optical bench temperature**

*LT*: Low Temperature(276K)

*HT1*: High Temperature1(290K)

*HT2*: High Temperature2(288K)

*HT3*: High Temperature3(299K)

*MT1*: Middle Temperature1(284K)

*MT2*: Middle Temperature2(283K)

*MT3*: Middle Temperature3(285K)

**CCD temperature:**

UV1,UV2: 254K

VIS1,VIS2: No temperature control

The wavelength shift and FWHM variety in different conditions is analyzed.

The pixel position corresponding to the emission peak of the mercury argon lamp is obtained by Gaussians fitting. The wavelength shifts of four channels are shown in Fig. 5.




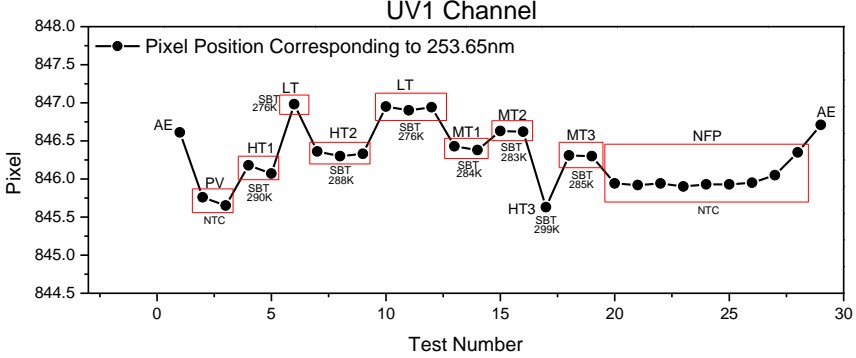

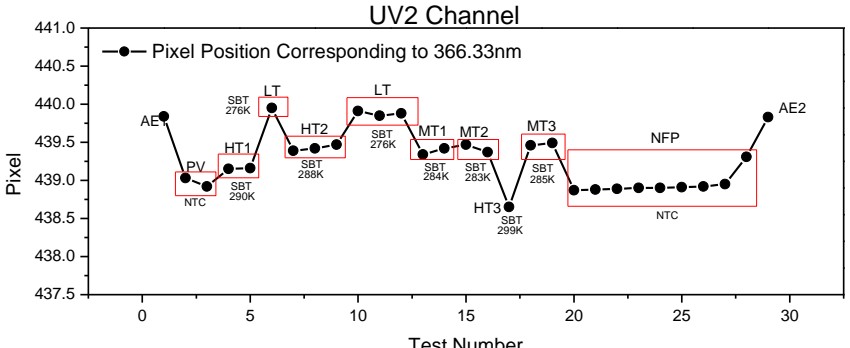

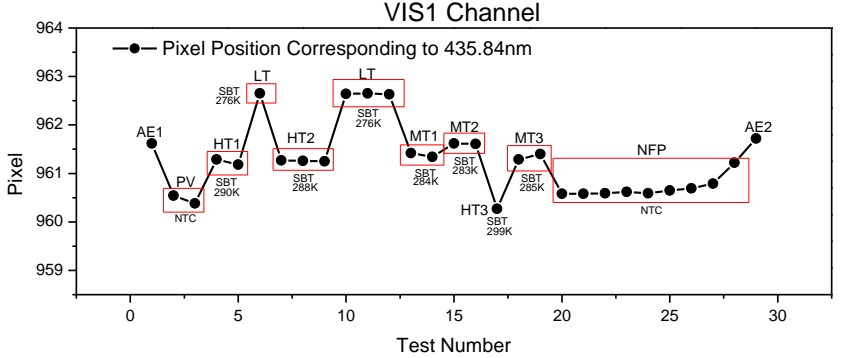

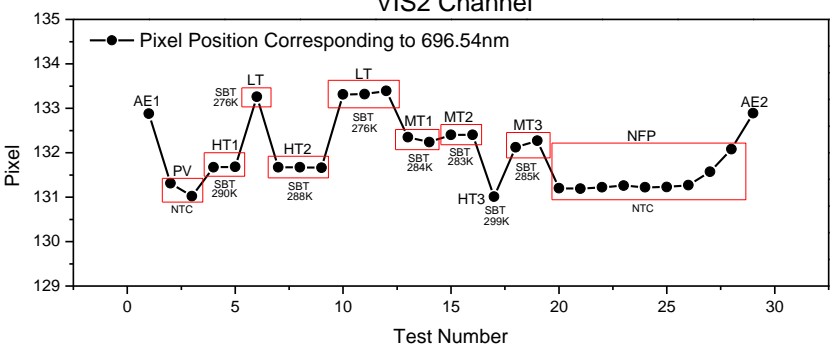




Fig.5. Wavelength shifts from atmospheric environment to vacuum: UV1/0.8pixel(about 0.06nm), UV2/0.8pixel(about 0.07nm), VIS1/1pixel(about 0.1nm), VIS2/1.5pixel(about 0.2nm); Wavelength shifts from HT1 to LT in vacuum: UV1:/1pixel(about 0.1nm), UV2/1pixel(about 0.1nm), VIS1/1.5pixel(about 0.2nm), VIS2/1.5pixel(about 0.2nm)

The wavelength shifts $\Delta\lambda$ are determined by

$$\Delta\lambda = \lambda_{Vac} - \lambda_{At} = (1 - 1/n)\lambda_{Vac}$$

where $\lambda_{Vac}$ and $\lambda_{At}$ is the wavelength in thermal-vacuum chamber and atmospheric environment, as the thermal-vacuum chamber pressure is smaller than atmospheric pressure, the atmospheric refractivity $n > 1$. The wavelength shift to long wave with the decrease of pressure($n$ becomes larger) and to short wave with the increase of pressure ($n$ becomes smaller) in thermal-vacuum chamber (see PV, NFP

results). And the wavelength shifts become larger with the increase of $\lambda_{Vac}$, the results show that the

shift is 0.06nm for 253.625nm and is 0.2nm for 696.54nm.

From the results, it also can be seen that the wavelength shifts change with the optical bench temperature in vacuum condition. The wavelength shift to long wave with the increase of optical bench temperature and to short wave with the decrease of optical bench temperature. The wavelength shift is

about 0.1nm for UV1, UV2 and is about 0.2nm for VIS1 and VIS2.

The FWHM of four channels are shown in Fig.6.

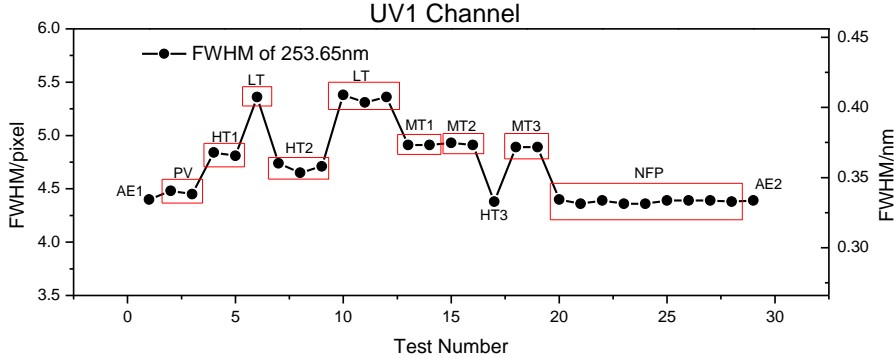



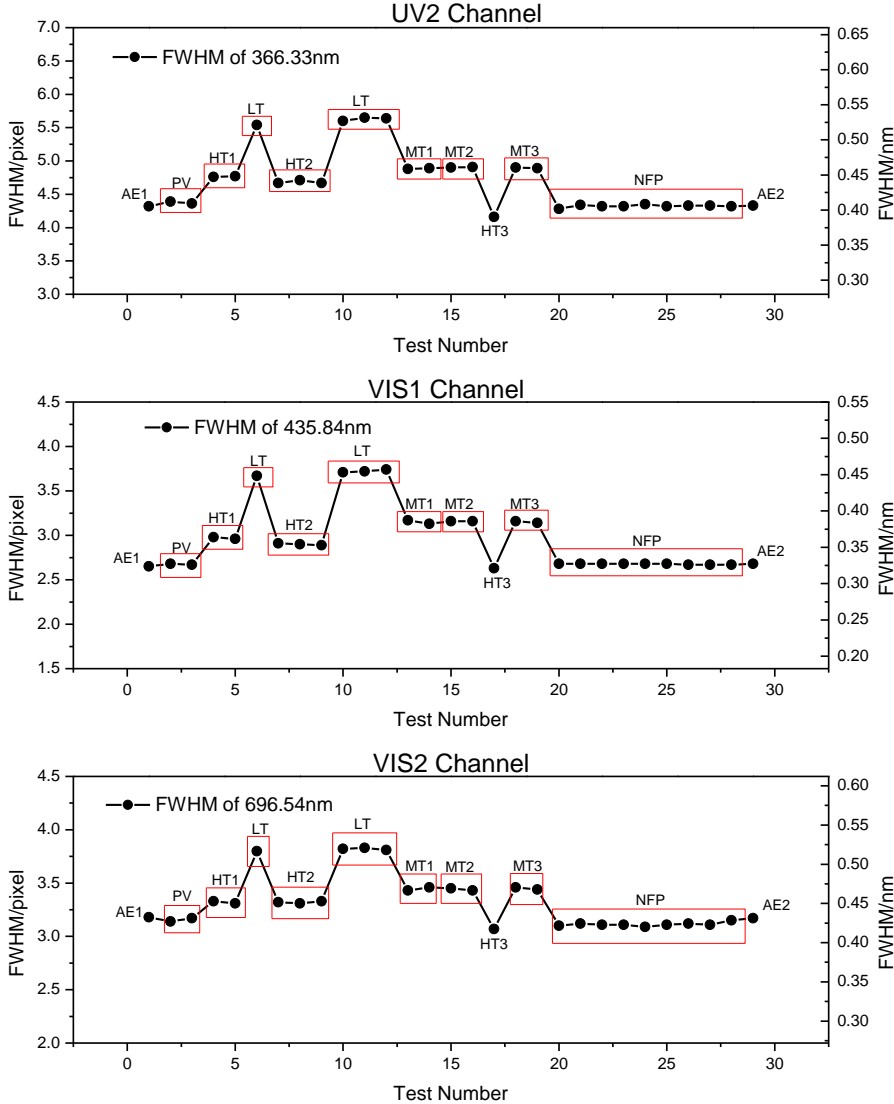


Fig.6. The FWHM results of thermal-vacuum test. The results show that, (1) the FWHM is basically the same in different pressure in thermal-vacuum chamber(see AE, PV, NFP results). (2) the FWHM become smaller with the increase of optical bench temperature in vacuum condition.

The FWHM changes with optical bench temperature see table 4.




Table 4. FWHM changes with optical bench temperature

| FWHM | Optical bench temperature/K | | | | | | |
|---|---|---|---|---|---|---|---|
| | 276 | 283 | 284 | 285 | 288 | 290 | 299 |
| UV1/ nm | 0.41 | 0.37 | 0.37 | 0.37 | 0.36 | 0.36 | 0.33 |
| UV2/ nm | 0.52 | 0.46 | 0.46 | 0.46 | 0.45 | 0.45 | 0.39 |
| VIS1/ nm | 0.45 | 0.39 | 0.39 | 0.39 | 0.36 | 0.36 | 0.32 |
| VIS2/ nm | 0.52 | 0.47 | 0.47 | 0.47 | 0.45 | 0.45 | 0.42 |

From the table 4, the optical bench temperature has a significant influence on the spectral resolution of the EMI. For example, the relative deviation of the spectral resolution between the optical bench temperature 276K and 299K is up to 25%. Therefore, the in-orbit optical bench temperature of the EMI
can be set up according to the FWHM results of the thermal-vacuum test.

**2.3 Spectral calibration in Solar calibration mode**

Spectral calibration in the Earth observation mode is introduced above. The calibration data in the Sun calibration mode shows that the same calibration results are obtained compared with the Earth
observation mode. We also get the solar spectrum from both mode on the ground, a optical fiber and a small telescope are used to introduce the direct sunlight to the Earth and Sun port. The solar spectrum in CFOV of the EMI(except the UV1) are shown in Fig.7, as the wavelength range in this channel is not visible on ground.

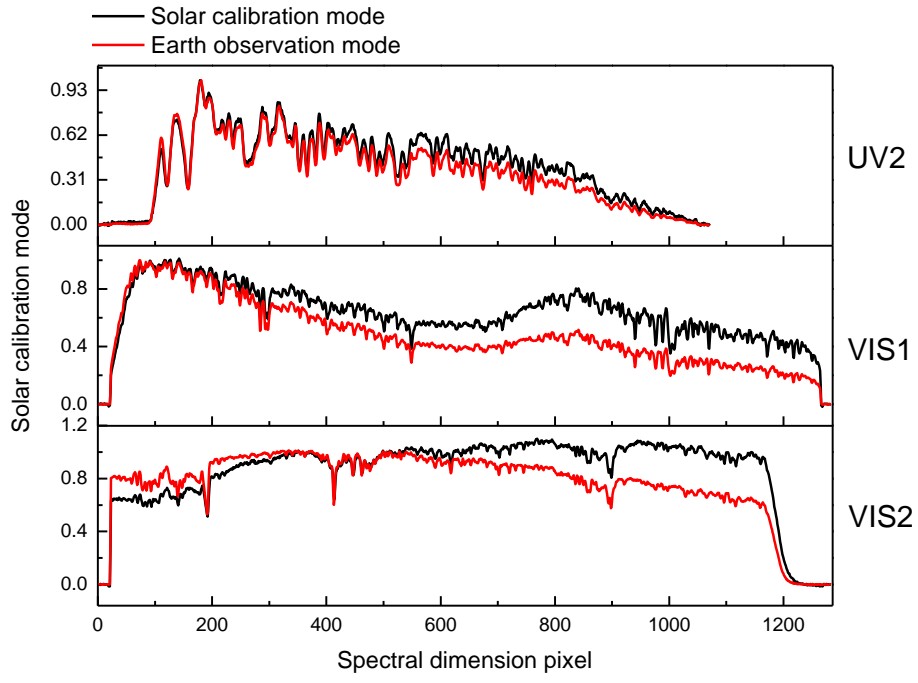



Fig.7. The solar spectrum obtained by EMI on the ground. The aluminum diffuser is used to observe the solar spectrum in the solar calibration mode(see Fig.1).

From Fig.7, it can be seen that the pixel corresponds to the same wavelength in two mode. The difference between the spectral shapes is due to the aluminum diffuser's spectral characteristics such as hemispheric reflectance and Bi-directional Reflectance Distribution Function (BRDF)[ *F. E. Nicodemus, et al., 1977, Kenneth J. Voss et al., 2000, Xuemin Jin et al., 2009*]. In addition, the spectral features of aluminum diffuser are introduced to the solar spectrum. The irradiance calibration of the Sun via space-borne diffuser is discussed latter.

## 3 Radiometric calibration

Radiometric calibration is carried out in the Earth observation and Solar calibration modes on the ground. In order to fulfill the requirements of in-orbit observation, several operating parameters are designed for the EMI instrument, such as three different integral times(0.5s, 1s , 2s) and 64 different gain values(0~63 with an interval of 1) corresponding to magnification of 0~5.8. The radiometric calibration is performed at different integral times, and the relationship between gain values and magnification is measured.

### 3.1 Radiometric calibration system

Integrating sphere and diffuse plate radiometric calibration system are used for EMI instrument. The integrating sphere system with tungsten halogen lamp is for the radiometric calibration of UV2, VIS1 and VIS2 channel. And the diffuse plate with a 1000W xenon lamp(Newport Xenon-6269) is for the UV1 channel(240~315nm), which produces sufficient ultraviolet output. The radiance of the radiometric calibration system is monitored by spectral radiometer: Ocean Optics MAYP11868(200~650nm) for diffuse plate system and USB2000(200~800nm) for integrating sphere system. Because it is not possible to illuminate the entire 114° instantaneously by the calibration system, the EMI instrument needs to rotate to complete the radiometric calibration.

The accuracy of the radiance directly determines the EMI radiometric calibration precision. Therefore, the spectral radiometers are also needed to be calibrated carefully. For this reason, the NIST-calibrated deuterium lamp(Newport) and 1000-W FEL quartz tungsten halogen lamp(OSRAM) are chosen to calibrate MAYP11868 and USB2000 separately. During calibration the lamp illuminate a stand diffuser plate, which convert the lamp irradiance to radiance to calibrate the spectral radiometer. The calibrated accuracy of the spectral radiometer is determined by three number of factors: the accuracy of the lamp irradiance standard, the accuracy of converting irradiance to radiance and the response accuracy of the spectral radiometer, which are discussed in detail below.

The accuracy of lamp irradiance is traced to NIST: deuterium lamp irradiance at 50cm is 3.16% in 210~350nm, FEL quartz tungsten halogen lamp irradiance at 50cm is 3.00%~2.40% in 250~400nm and 2.40%~1.60% in 400~800nm.

The method of converting irradiance to radiance is given by



$$L_{rad} = E_{lamp-irrad} \cdot (\frac{l_{lamp-plate}}{l_{50cm}})^2 \cdot BRDF_{std-plate}$$

Where $L_{rad}$ is the radiance converting form the lamp irradiance $E_{lamp-irrad}$ at $l_{lamp-plate}$, which is 50cm

for the spectral radiometer calibration, $l_{50cm} = 50cm$, the stand diffuser plate $BRDF_{std-plate}$ is close to

$\frac{1}{\pi}(sr^{-1})$, with the accuracy of 1.25%. The distance between the stand diffuser plate and the lamp is 500

$\pm$1mm.

A optical fiber and a small telescope are used by spectral radiometer to observe the stand diffuser plate at an angle of $40°$ . One hundred observed data is obtained by the spectral radiometer, the accuracy of MAYP11868 response stability is better than 0.80%, and the accuracy of USB2000 response stability is better than 0.50%. In practice, the radiance monitored by the spectral radiometer is

usually different from the radiance of the diffuser plate, therefore, the spectral radiometer needs to work in the linear response region. Five different radiance levels are observed by the spectral radiometer to determine the accuracy of the response linearity, the results show that the accuracy of MAYP11868 response linearity is better than 1.20%, and the accuracy of USB2000 response linearity is better than 1.10%.


Table 5. Calibrated accuracy of the spectral radiometer

| Uncertainty/% | MAYP11868 (210nm~350nm) | USB2000 (250nm~400nm/400nm~800nm) |
|---|---|---|
| **Lamp irradiance standard** | *3.16* | *3.00~2.40/2.40~1.60* |
| **Converting** (Irradiance to radiance) | *1.27* | *1.27* |
| **Spectral radiometer** | *1.44* | *1.21* |
| **Total** | *3.70* | *3.48~3.00/3.00%~2.38* |

For the diffuse plate radiometric calibration system, 1000W xenon lamp illuminate the same stand diffuse plate discussed above to produce near uniform surface light source, which is also produced at the integrating sphere opening by introducing the halogen tungsten lamp light to the sphere via a round

pipe. The two radiometric calibration systems have their own highly stabilized power supply. The accuracy of surface light source includes the surface uniformity and stability. The radiometric accuracy of the calibration system is shown in table 6.





Table 6. Radiometric accuracy of the calibration system

| Uncertainty/% | Diffuse plate system (210nm~350nm) | Integrating sphere (250nm~400nm/400nm~800nm) |
|---|---|---|
| Surface uniformity | < 2.00 | < 2.00 |
| Surface stability | < 0.10 | < 0.10 |
| Spectral radiometer | 3.70 | 3.48~3.00/3.00~2.38 |
| Total | < 4.21 | < 4.02~3.61/3.61~3.11 |

### 3.2 Radiance calibration

The data $N_{signal}$ collected by EMI including dark signal $N_{dark}$ and light signal $N_{light}$ is given by

$$N_{signal} = N_{dark} + N_{light}$$

where $N_{dark}, N_{light} \propto T_{time}, G_{gain}$, the integral time $T_{time}$ can be set as 0.5s, 1s , 2s, the gain $G_{gain}$ can be set form 0 to 63 with the interval of 1.

In order to obtain an approximate dark correction and to widely remove the dark-current-induced spectral structures, the mean dark spectra is subtracted[*Birgre Bohn et al., 2017*]. The dark signal and light signal are discussed separately below.

**Dark signal**

EV2-CCD4720 and EV2-CCD5530 are adopted for UV and VIS channel separately. As the weak ultraviolet band of the atmospheric light, the two UV channel CCDs are cooled to -20℃ to reduce dark signal.

The dark signal for each pixel is composed of an electronic offset $N_{offset}$ and dark noise $N_{noise}$

$$N_{dark} = N_{offset} + N_{noise}$$

The offset is fairly const, but dark noise is a thermally induced dark-current signal increasing with temperature and integration time[*Evelyn Jakel et al., 2007*]. Therefore, dark signal measurement should be conducted frequently to update the dark data. The dark signal under different integral time is shown in fig.5, which take UV2 channel and VIS1channel for example.



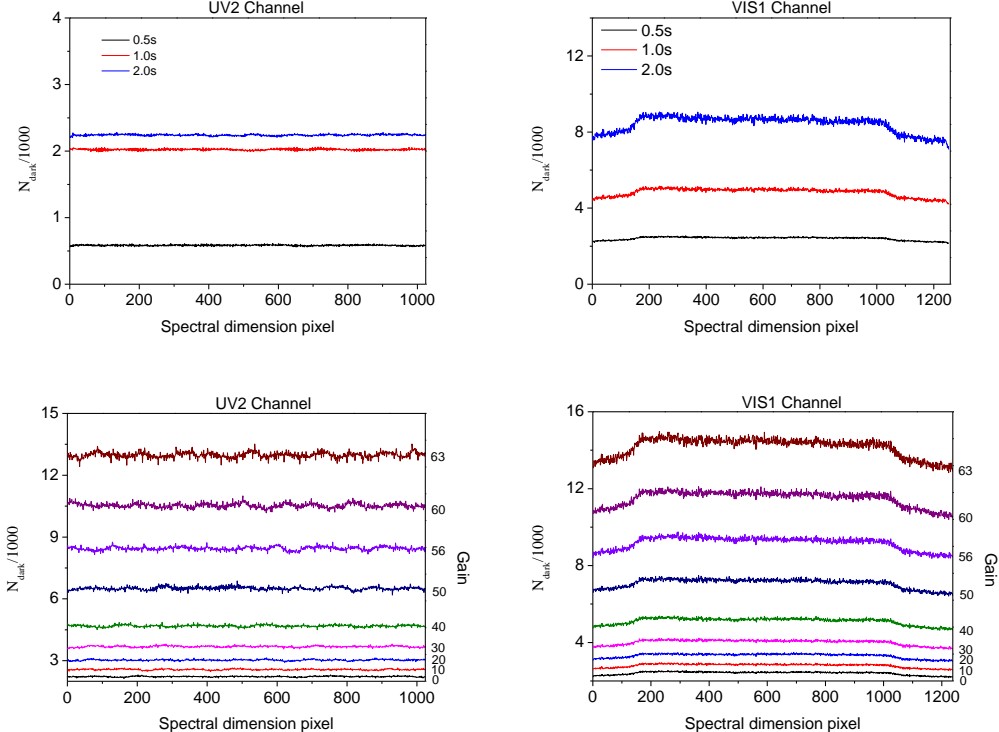

Fig.8. Top: Dark signal under different integral time. Bottom: Dark signal under different gain. The gain is set to 0,10,20,30,40,50,56,60,63.

From Fig.8, the small spectral structure in dark signal is caused by dark noise, which could influence the measured data, especially under weak-light conditions. The dark noise can be get by deriving standard deviations of repeated dark measurements, and can be reduced by averaging the repeated dark data. The dark spectra is recorded for each orbit when EMI is in orbit, and then the dark spectra under the same work conditions are averaged to correct the observation spectra.

**Light signal**

The output radiance level of the radiometric calibration system is determined by the xenon lamp output power for diffuse plate system, and is determined by the introduction of the light for integrating sphere system. For UV1 channel, the EMI instrument views the standard diffuse plate at an angle of 45.0° and at a distance of 50.0cm, about 13° viewing angle of EMI can be illuminated once, so the instrument has to be rotate in 9 steps to complete the entire 114°. For UV2, VIS1, VIS2 channels, EMI views the integrating sphere opening at a distance of 40.0cm, about 11° can be illuminated once, and 11 steps are needed to complete the radiance calibration.

The dark signal is deducted from the radiance calibration data firstly. One radiance level of the radiance calibration systems and the corresponding response of the EMI instrument are shown in Fig.9.





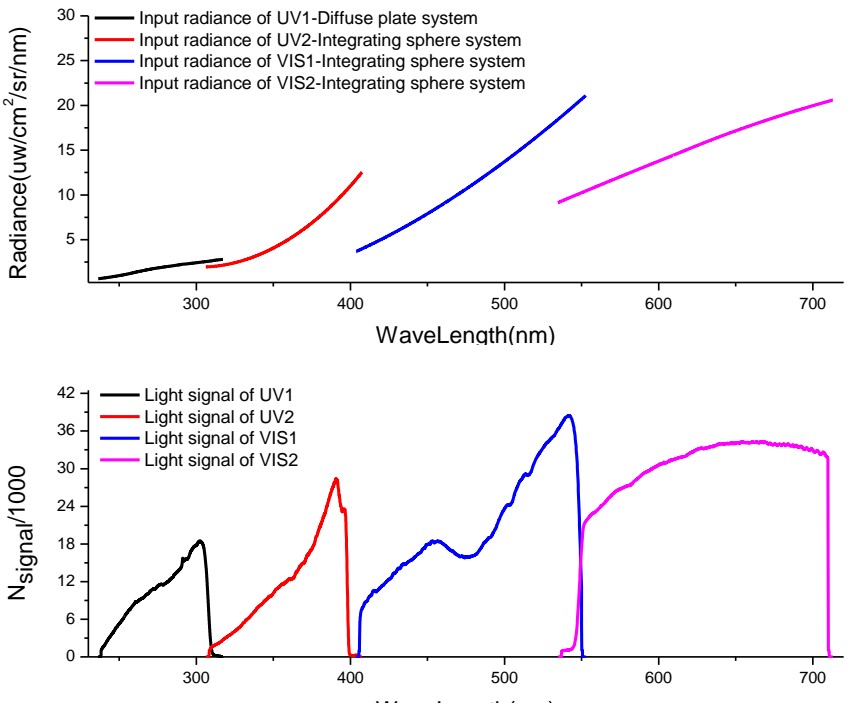

Fig.9. The upper panel presents one radiance level of diffuse plate and integrating sphere system. The lower panel presents the EMI response to the radiance, the dark signal is subtracted from the response .The work parameters of UV1, UV2, VIS1, VIS2 are : the integral time 2s,1s,1s,1s, and gain 0, 0, 0, 0.

      From Fig.9, there is an overlap band at each end of the channels, which is due to the optical
features of the color separation filters. In addition, the response in the wavelength range 460~480nm of VIS1 channel become lower because a filter of this range is placed in front of the slit 11, the purpose is to make sure that the detectors are not saturated in the case of clouds.

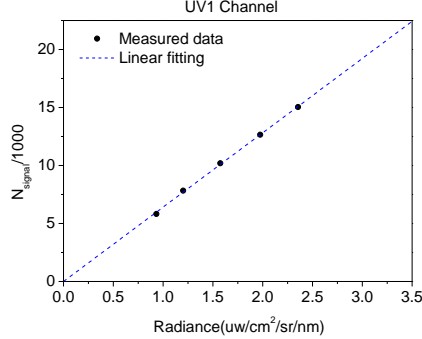

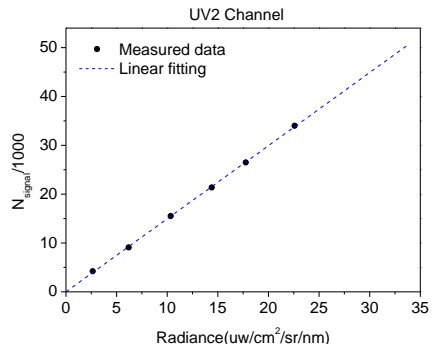



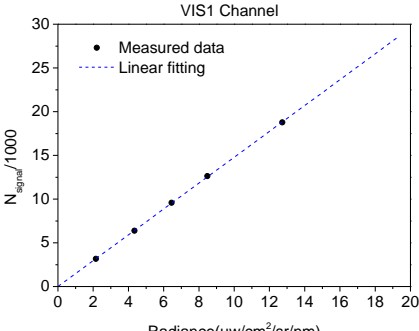
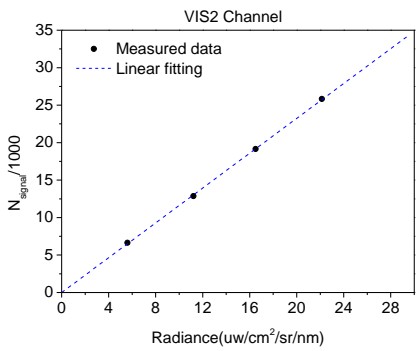

Fig.10. Linear response of the EMI, the signal is corrected by dark signal. Note that, there is the non-linear response region in the very low light signal(equal to the dark signal) and high light signal(saturation light signal) conditions. The integral time, CCD readout and gain are set up to ensure the EMI works in the linear response region.

       Base on the linear response of the EMI, the radiance calibration model is

$L_{radiance} = \alpha \cdot N_{Light}$

$L_{radiance}$ is the radiance at the EMI entrance pupil, $\alpha$ is the radiance response coefficient.

The theoretical relation between gain $f_{gain}$ and magnification $f_{magn}$ is determined by

$$f_{magn} = \frac{5.8}{1 + 4.8 \cdot (63 - f_{gain})/63}$$

       The light signal under different gain is shown in fig.11, which take UV2 and VIS1channel for
example.

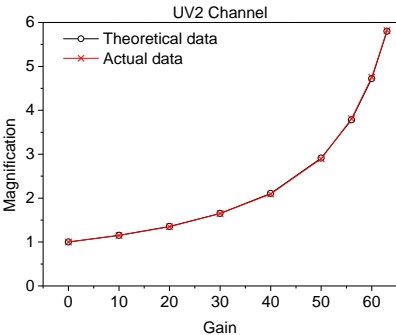
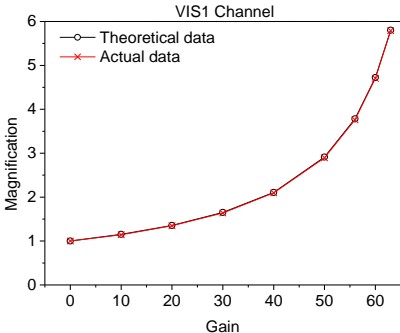

Fig.11. The relation between gain and magnification, the results show that the relative deviation between theoretical and actual data is better than 1.0%. In application, the magnification can be obtained from the theoretical relation.



The overall accuracy of the radiance calibration is mainly determined by the accuracy of the radiance calibration system, by the response non-linearity and by the response non-stability of the EMI. The accuracy of the diffuser plate system and integrating system is shown in table. The response non-linearity can be calculated by the data in Fig.8, and the results are 1.13%(UV1), 1.04%(UV2), 1.07%(VIS1) and 1.00%(VIS2). The response non-stability is obtained by 1000 repeated spectra of the

EMI, and the results are 1.21%(UV1), 1.26%(UV2), 1.12%(VIS1) and 1.14%(VIS2). The accuracy of the conversion of different gains should be consider in the case of the light signal corrected by the gain. The final accuracy of the radiance calibration is shown in table 7.

Table7. Radiance calibration accuracy

| Channel | Accuracy(%) | |
| --- | --- | --- |
| | No gain corrected | Gain corrected |
| UV1 | 4.53 | 4.64 |
| UV2 | 4.52 | 4.63 |
| VIS1 | 4.31 | 4.43 |
| VIS2 | 4.30 | 4.42 |

## 3.3 Irradiance calibration

The solar irradiance is calibrated mostly via the onboard diffusers[*S.Noel et al., 2006, Xiaoxiong Xiong et al., 2009*]. The irradiance calibration depends on the incident angles on the onboard diffusers of the EMI. The azimuth angle varies slowly throughout the year from about 16° to 28° around the nominal value of 22°, the elevation angle varies from +4° to -4° around the nominal value of 11°. The elevation

angle change originates from the satellite orbital movement. About 75 images are obtained during a solar observation sequence of 150s, and each individual image needs to be corrected for the radiometric goniometry.

$$DN_{\alpha_0,\beta_0} = DN_{\alpha,\beta} \cdot f_{\alpha,\beta}$$

Where $DN_{\alpha_0,\beta_0}$ is the image at the nominal azimuth angle $\alpha_0$ and elevation angle $\beta_0$, which is

corrected from the $DN_{\alpha,\beta}$ with the goniometry correction factor $f_{\alpha,\beta}$. And the corrected images are averaged to improve the SNR. The irradiance calibration model of the EMI is



$$I_{Sun} = [\frac{1}{n}\sum_{i=1}^{n}(DN_{\alpha,\beta} \cdot f_{\alpha,\beta})_i] \cdot \sigma_{\alpha_0,\beta_0}$$

where $n = 75$, $\sigma_{\alpha_0,\beta_0}$ is the irradiance response coefficient. The goniometry correction factor and

irradiance response coefficient of the EMI are calibrated on the ground. A light source has a beam

divergence that is comparable to the sun, which is rotated to cover the azimuth and elevation angle ranges. The goniometry correction factors are shown in Fig , which are by definition 1.00 for the nominal azimuth and elevation angles.

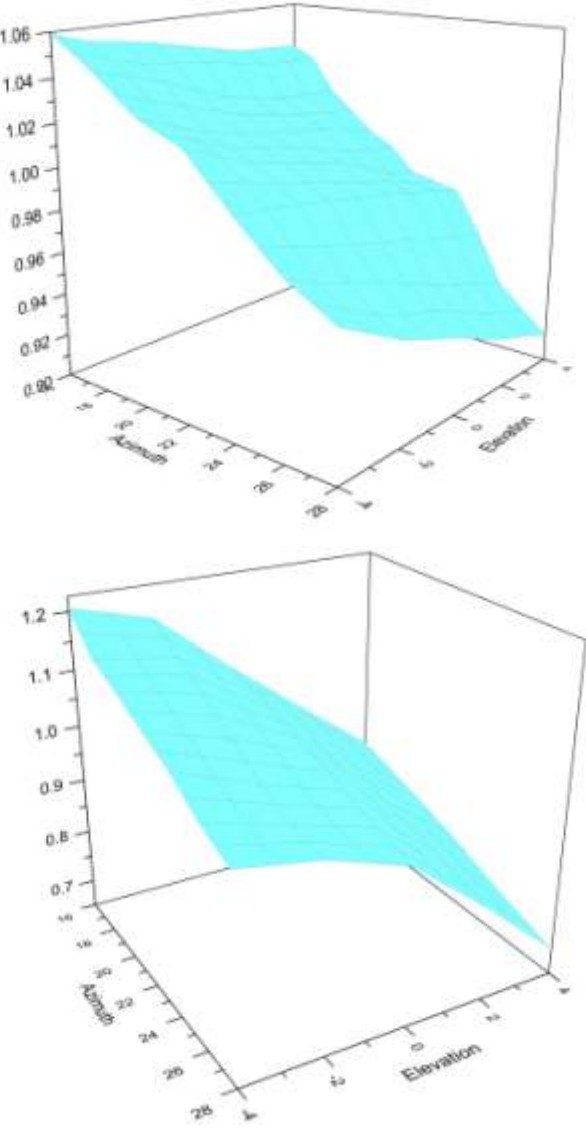



Fig.12. Goniometry correction factors for the aluminum diffuser(upper panel) and quartz volume diffuser(lower panel) for the center field of view.

The NIST-calibrated 1000W FEL quartz tungsten halogen lamp is used for the absolute irradiance calibration at the nominal azimuth and elevation angles. And the irradiance response coefficient $\sigma_{\alpha_0,\beta_0}$ is obtained for the irradiance calibration model of the EMI.

It was found that aluminum diffusers adopted by the SCIAMACHY project introduce spectral structures in the Sun reference spectrum[*C.E. Sioris et al., 2004*]. These structures are comparable to trace gas absorption features. They may interfere with DOAS-based retrieval of trace gases hence affecting the accuracy of the retrieved column densities[*A.Richter, et al., 2001,2002, Courreges-Lacoste et al., 2004*]. As the QVD introduce considerably less structure than aluminum diffuser, the EMI used it to provide the

solar reference spectrum once per day. The aluminum diffuser is mainly used for radiometric calibration purpose, which is performed once monthly.

In addition, EMI works in low Earth orbit (LEO) with the orbit altitude of 708 km. The critical space environment will affect the performance of materials and components in LEO[*Samuel F. Pellicori,2014*]. Such as atomic oxygen (AO)[ *Bruce A. Banks et al., 2008*], Solar UV and the energetic

protons trapped in the inner Van Allen belt. Space radiation exposure effects on onboard diffusers have been tested and discussed by[*MinJie Zhao, et al., 2015*].

## 4 Signal to noise ratio

The EMI is needed to meet the signal to noise ratio(SNR) requirements for dark scenes(especially in the UV bands)[ *Johan de Vries et al., 2009*], to ensure the accuracy of retrieved results. A SNR model is

introduced, which is in good agreement with the experimental result. And the EMI in-orbit SNR is estimated by using the SNR model and MODTRAN. The SNR estimation for advanced hyperspectral space instrument is discussed by[*Andreas Eckardt et al., 2005, Lang Junwei et al., 2013* ].

The electrons generated by a signal pixel can be calculated by

$$s_e = \frac{\pi}{4}(\frac{D}{f})^2 \cdot \tau(\lambda) \cdot L(\lambda)\frac{A_d t_{int}\lambda}{hc}\eta(\lambda)\Delta\lambda$$

where D/f is relative aperture of optics, $h$ is the Plank constant, $c$ is the light speed, $\tau(\lambda)$ is transmission of optics, $L(\lambda)$ is sensor input radiance in $uw/cm^2/sr/nm$, $\Delta\lambda$ is spectral bandwidth of a single spectral line, $A_d$ is pixel area, $t_{int}$ is integration time, $\eta(\lambda)$ is Quantum efficiency of CCD.

The main part of the total noise is the shot/photon noise generated by the incident radiation. The shot/photon noise can be described by the Poisson distribution, and can be calculated as

$$\delta_{shot} = \sqrt{S_e}$$

The other noise include dark noise $\delta_{dark}$ and read out noise of the CCD $\delta_{read}$. Generally the SNR can be calculated by

$$SNR = \frac{S_e}{\sqrt{\delta_{shot}^2 + \delta_{dark}^2 + \delta_{read}^2}}$$

The SNR can be improved by pixel binning,

$$SNR = MS_e / \sqrt{MS_e + M\delta_{dark}^2 + \sigma_{read}^2}$$

where $M$ is the binning factor, see table 1.

The output digital number of a signal pixel is obtained by the conversion factor $f$ of the CCD:

$$DN = f \cdot S_e$$

For the SNR model of the EMI, it is impossible to measure the signal and noise separately. In
practice, one way is to adopt the mean value of the repeat DNs as the signal and to adopt the standard deviation of the repeat DNs as the noise. In this case, N repeated measured spectra of EMI is recorded by observing the uniform-stable light source of the calibration system. And the measured SNR is calculated by

$$SNR = \frac{\overline{DN}}{\sqrt{\frac{\sum_{i}^{N}(DN_i - \overline{DN})}{N-1}}}$$

The offset is deducted from the DNs. Fig.13 show the simulation and measured SNR results of VIS1 at the input sensor radiance.





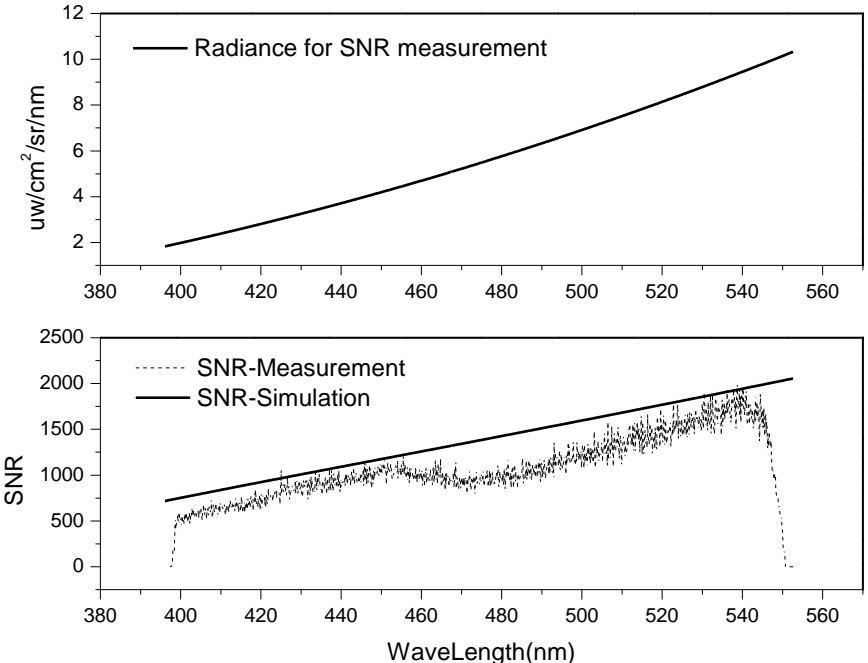

Fig.13. The upper panel presents the radiance of the integrating sphere system for SNR measurement of VIS1 in lab. The lower panel presents the results of measured SNR(solid line) and simulation result(dotted line) for the radiance of the upper panel, with the integration time of 2s and the binning factor of 4. The measured SNR in the wavelength range 460~500nm become lower because a filter of this range is placed in front of the slit 11, the purpose is to make sure that the detectors are not saturated in the case of clouds. And there is an overlap band at the end of the channel, which is due to the optical features of the color separation filter. In addition, there are 24 dark pixels at the end of the channel with the measured SNR is about zero.

For the measured SNR, 100 repeated measured spectra of EMI is recorded by observing the integrating sphere system with the integration time of 2s and the binning factor of 4. The offset signal is deducted from the spectra using the equation?. For the simulation SNR, the F-number of EMI optics $F\# = 3.2$, the spectral width of VIS1 $\Delta\lambda = 0.12nm$, the area of a singl pixel $A_d = 22.5 \times 22.5(um^2)$,the integration time $t_{int} = 2s$, and the binning factor $M = 4$. Fig show that the measured SNR is lower than the simulation SNR, the possible reasons are the non-uniformity and non-stability of the light source and the pixel response non-uniformity(PRNU) for the binning pixels. But the results also show that it is a good choice to estimate the EMI in-orbit SNR using the SNR model.

The simulation EMI in-orbit SNR of the UV2, VIS1 and VIS2 are shown in Fig. 14. The in-orbit





SNR of this channel is not estimated as the solar light in the band of the UV1(240-310nm) is absorbed by the atmosphere.

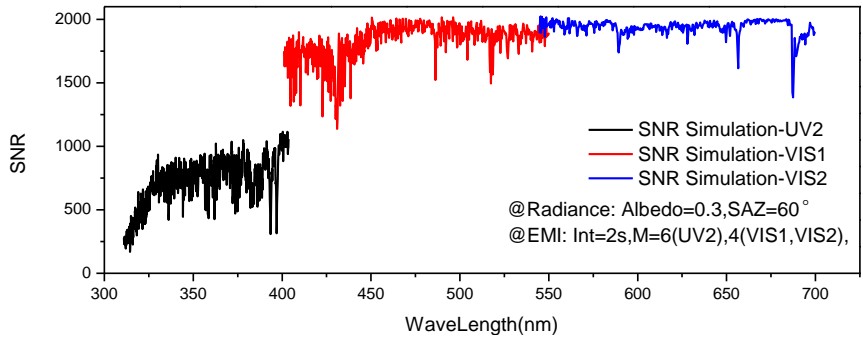

Fig.14. The simulation EMI in-orbit SNR of UV2, VIS1 and VIS2. The input radiance for the SNR model is obtained by MODTRAN with the albedo of 0.3 and whit the sun zenith of $60°$ . The EMI simulation SNR with the integration time of 2s, the binning factor of 6 for UV2 channels and 4 for VIS channels, the spectral bandwidth of a pixel of 0.09nm for UV2, 0.12nm for VIS1 and 0.13nm for VIS2.

The in-orbit radiance obtained by MODTRAN for an albedo of 0.3 at $60°$ sun zenith is used for the
simulation EMI in-orbit SNR. The in-orbit SNR of the UV2 is about 700, the VIS1 is about 1800 and the VIS2 is about 2000. In the condition of the dark scenes, the SNR can be improved by the increase of the binning factor.

## 5 Conclusions

The spectral and radiometric response performance of the EMI is obtained by the preflight calibration.
At the same time, the obtained calibration key data is used for the L1b processor. After launch, the EMI in-orbit performance may change due to the vibration of the launching and the change of the environment conditions. Therefore, the EMI in-orbit calibration is performed in order to verify preflight calibration and ensure calibration accuracy. For the EMI, the in-orbit wavelength calibration is performed by use of the Fraunhofer lines in the sun spectra and Earth spectra. The in-orbit radiometric
calibration is performed by observe the sun via the onboard diffusers. During the EMI flight, the low Earth orbit space environment such as atomic oxygen, the solar UV and energetic protons will affect the EMI response performance, the aluminum diffuser and the quartz tungsten halogen white light source(6V, 10W) are used to monitor the disintegration of the EMI.

**Acknowledgements.**
This research was supported by grant from National Natural Science Foundation of China (41705016).



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
