# Peer review of "Preflight Calibration of the Chinese Environmental Trace Gases Monitoring"

_Atmospheric Measurement Techniques, 2018_

## Referee Comment (RC1) · Anonymous Referee #2 · 15 May 2018

The manuscript entitled "Preflight Calibration of the Chinese Environmental Trace Gases Monitoring Instrument (EMI)" by Zhao et al. describes the method of the pre-flight wavelength and radiometric calibration efforts for the EMI instrument. Moreover, it provides an estimate of the expected, on-orbit signal to noise ratio for one particular solar zenith angle.

In my opinion, this manuscript provides valuable information to the community, but requires careful modifications before it is published. My detailed comments are:

(1) There are several editorial and vocabulary issues, possibly due to a language barrier, that make the manuscript hard to read and sometimes result in the incorrect meaning. Please proof-read the manuscript carefully. Several examples are listed in the following:

[Figure]

a. "integral time" should be "integration time"

b. The symbol "∼" is used throughout the manuscript to describe "from/to" intervals or ranges. The correct symbol to use is "-".

c. The word "data" is used to describe "measurements". For example, "…determined by 20 spectral response data…" should be modified to "determined using 20 spectral response measurements…". Similarly, "One hundred observed data is obtained…" should be modified to read: "One hundred measurements were obtained…"

d. "…the spectral response function is better than 0.03nm." should be modified to "the full width at half maximum (FWHM) of the instrumental line shape function is less than 0.03nm."

e. Throughout the manuscript, the abbreviation "FWHM" is used for the FWHM of the instrumental line shape function (ILS). Whenever it is used, it has to be made clear that it describes the ILS and not the width of some other function.

f. In section 3, gain steps between 0-63 are introduced which result in different gain values within the CCD readout electronics (A/D converter). However, the word "Gain" is used for the digital gain steps and the word "magnification" is used for the actual gain value. I strongly encourage the authors to describe the values 0-63 as "gain steps" (or something similar) and the factor with which the raw signal is multiplied as "gain" or "gain value". In the community, the word "magnification" is almost exclusively used for optical magnifications, which can result in confusion here. Please do not use "magnification" in this context.

g. The words "accuracy" and "precision" (and sometimes "non-stability" or "variety") are sometimes used interchangeably and often wrongly in this manuscript. Please familiarize yourself with the different meanings of accuracy and precision and use them appropriately. Do not use non-stability or variety.

h. I assume the CCD names are "e2v…" not "EV2…"

i. The dark signal is incorrectly defined in line 288. The common way to define the signal that is obtained when no photons enter the instrument is to add the "bias value" and the "dark signal", where the dark signal is the dark current multiplied by the integration time. The dark noise is typically the noise component that is caused by this dark signal, in this case, the shot noise of the dark signal.

j. Figure number is missing in line 366.

k. The unit Watt is typically abbreviated with a capital "W", not a lower case "w".

l. Equation number is missing in line 428. In fact, the equations are not numbered at all. Please assign equation numbers to all equations.

m. Please use the greek letter $\mu$ to indicate thousandths not the letter u.

n. Figure number is missing in line 430.

(2) It is not sufficiently clear what the wavelength shifts shown in Figure 3 are. Do they represent an additional offset that is included in the polynomial function which is determined for the center?

(3) The manuscript states that the CCDs for the visible channels do not have any temperature control. Since the dark current depends strongly on the CCD temperature, it would be very helpful to quote the expected temperature variations of these detectors throughout the orbit and as a function of orbit beta angle. In addition, it would be helpful to refer to the strategy of periodic dark measurements at this point, so the reader understands how this potential problem is mitigated.

(4) The authors state: "The offset is fairly const, . . ." I believe they mean "The bias value is constant,. . ." This is generally a good assumption for well-designed electronics. Have the authors quantified the precision of the bias values?

(5) I do not understand the traces in the top two panels of Figure 8. For a constant dark current and a constant bias value, the difference between the measurements with 0.5s

and 1.0s integration time should be half of the difference between the measurements with 1.0s and 2.0s integration time. Please explain.

(6) A reference for MODTRAN should be included.

(7) The denominator of the equation on line 414 should be the standard deviation. Thus, the term in the sum needs to be squared. I assume that the actual calculations were performed correctly.

(8) The authors state that the measured SNR in figure 13 is departing from the simulation between 460-500nm due to lower transmittance of the instrument (filter) in this range. However, if the equation in line 394 includes the proper transmission function, this effect should be included in the simulation. Please explain.

(9) It is not clear to me how the PRNU can provide a significant contribution to the lower than expected SNR, unless it is varying in time (line 432). Please explain.

(10) If I understand correctly, the pre-flight, radiometric calibration of EMI was not conducted under flight-like vacuum and possibly thermal conditions. If this is the case, please address in more detail how the in-flight calibration will be used to accomplish absolute radiometric calibration of the flight data.

(11) Finally, while the manuscript shows the performance of the instrument on the ground, the reader is not told what the actual performance requirements are. Presumably, the instrument performance requirements are driven by the scientific objectives. Comparing the measured/estimated performance (e.g. SNR) with the mission requirements would make the conclusion much stronger.

---

## Author Comment (AC1) · 22 Jun 2018

We would like to thank you for the insightful comments. Our responses to the comments are given below.

General comments:

The manuscript entitled "Preflight Calibration of the Chinese Environmental Trace Gases Monitoring Instrument (EMI)" by Zhao et al. describes the method of the preflight wavelength and radiometric calibration efforts for the EMI instrument. Moreover, it provides an estimate of the expected, on-orbit signal to noise ratio for one particular solar zenith angle. In my opinion, this manuscript provides valuable information to the community, but requires careful modifications before it is published. My detailed comments are:

(1) There are several editorial and vocabulary issues, possibly due to a language barrier, that make the manuscript hard to read and sometimes result in the incorrect meaning. Please proof-read the manuscript carefully. Several examples are listed in the following:

a. "integral time" should be "integration time"

b. The symbol "∼" is used throughout the manuscript to describe "from/to" intervals or ranges. The correct symbol to use is "-".

c. The word "data" is used to describe "measurements". For example, ": : :determined by 20 spectral response data: : :" should be modified to "determined using 20 spectral response measurements: : :". Similarly, "One hundred observed data is obtained: : :" should be modified to read: "One hundred measurements were obtained: : :"

d. ": : :the spectral response function is better than 0.03nm." should be modified to "the full width at half maximum (FWHM) of the instrumental line shape function is less than 0.03nm."

e. Throughout the manuscript, the abbreviation "FWHM" is used for the FWHM of the instrumental line shape function (ILS). Whenever it is used, it has to be made clear that it describes the ILS and not the width of some other function.

f. In section 3, gain steps between 0-63 are introduced which result in different gain values within the CCD readout electronics (A/D converter). However, the word "Gain" is used for the digital gain steps and the word "magnification" is used for the actual gain value. I strongly encourage the authors to describe the values 0-63 as "gain steps" (or something similar) and the factor with which the raw signal is multiplied as "gain" or "gain value". In the community, the word "magnification" is almost exclusively used for optical magnifications, which can result in confusion here. Please

do not use "magnification" in this context.

g. The words "accuracy" and "precision" (and sometimes "non-stability" or "variety") are sometimes used interchangeably and often wrongly in this manuscript. Please familiarize yourself with the different meanings of accuracy and precision and use them appropriately. Do not use non-stability or variety.

h. I assume the CCD names are "e2v: : :" not "EV2: : :"

i. The dark signal is incorrectly defined in line 288. The common way to define the signal that is obtained when no photons enter the instrument is to add the "bias value" and the "dark signal", where the dark signal is the dark current multiplied by the integration time. The dark noise is typically the noise component that is caused by this dark signal, in this case, the shot noise of the dark signal.

j. Figure number is missing in line 366.

k. The unit Watt is typically abbreviated with a capital "W", not a lower case "w".

l. Equation number is missing in line 428. In fact, the equations are not numbered at all. Please assign equation numbers to all equations.

m. Please use the greek letter μ to indicate thousandths not the letter u.

n. Figure number is missing in line 430.

*Response*:

Many thanks for the careful and professional commenting. Firstly, the comments a-n have been corrected in the paper. Secondly, the paper is carefully modified.

(2) It is not sufficiently clear what the wavelength shifts shown in Figure 3 are. Do they represent an additional offset that is included in the polynomial function which is determined for the center?

*Response:*

The wavelength shifts in Figure 3 are measured by the tunable laser in the spatial dimension with the interval of 5°.

[Figure]

The wavelength (pixel) shift enlarges from the CFOV to the edge FOV. The UV1, UV2, VIS1, and VIS2 wavelength (pixel) shifts of the edge FOV are 1.12 nm (14

pixels), 0.9 nm (10 pixels), 1.2 nm (10 pixels), and 1.3 nm (10 pixels), correspondingly. For the L1b processor of the EMI,the spectral smile effect will be calibrated using a spectrum-matching technique.

(3) The manuscript states that the CCDs for the visible channels do not have any temperature control. Since the dark current depends strongly on the CCD temperature, it would be very helpful to quote the expected temperature variations of these detectors throughout the orbit and as a function of orbit beta angle. In addition, it would be helpful to refer to the strategy of periodic dark measurements at this point, so the reader understands how this potential problem is mitigated.

*Response:*

The CCDs for the visible channels do not have independent temperature control, but they work in a constant temperature environment. The temperature is similar to that in the visible spectrometer, which has temperature control. Thus, the change of CCD temperature is not a problem.

(4) The authors state: "The offset is fairly const, *∶∶∶*" I believe they mean "The bias value is constant,*∶∶∶*" This is generally a good assumption for well-designed electronics. Have the authors quantified the precision of the bias values?

*Response:*
The read-out register within the CCD has an excess of 16 blank pixels, which can be used to measure the electronic offset on the ground. The measurements show that the offset is not constant but drifts with time (about 0.5%). Therefore, the electronic offset is obtained per measurement frame in-orbit, and the electronic offset correction is implemented in the L1b data processor.

(5) I do not understand the traces in the top two panels of Figure 8. For a constant dark current and a constant bias value, the difference between the measurements with 0.5s and 1.0s integration time should be half of the difference between the measurements with 1.0s and 2.0s integration time. Please explain.

*Response:*
The pixels in the readout register cannot be used to accomplish the binning due to the full well limitation. In this case, the pixel binning is accomplished in the Field Programming Gate Array. Fast readout frequency is needed for the process. The fast readout frequency leads to signal distortion. Therefore, the difference between the measurements with 0.5 and 1.0 s integration times is not half of the difference between the measurements with 1.0 and 2.0 s integration times. Based on the signal distortion, we have obtained absolute radiance calibration key data at different integration time on the ground. The calibration key data are used for the L1b data processor.

(6) A reference for MODTRAN should be included

*Response:*
A reference for MODTRAN have been included in the paper.

(7) The denominator of the equation on line 414 should be the standard deviation. Thus, the term in the sum needs to be squared. I assume that the actual calculations were performed correctly.

*Response:*
The equation in the paper has been corrected. We have confirmed that the actual calculations were performed correctly.

(8) The authors state that the measured SNR in figure 13 is departing from the simulation between 460-500nm due to lower transmittance of the instrument (filter) in this range. However, if the equation in line 394 includes the proper transmission function, this effect should be included in the simulation. Please explain.

*Response:*
The equation in line 394 includes the proper transmission function. But for the SNR-simulation, the transmittance of the filter is not included as we want to analyze the effect of the filter on SNR. The simulation SNR included the effect of the filter is shown in following figure.

[Figure]

(9) It is not clear to me how the PRNU can provide a significant contribution to the lower than expected SNR, unless it is varying in time (line 432). Please explain.

*Response:*

The PRNU is not varying during the SNR measurement, and will not provide a significant contribution to the lower than expected SNR. There are two main factors: the light source for the SNR measurement and the pixel response of the EMI. The PRNU has been corrected in the paper.

(10) If I understand correctly, the pre-flight, radiometric calibration of EMI was not conducted under flight-like vacuum and possibly thermal conditions. If this is the case, please address in more detail how the in-flight calibration will be used to accomplish absolute radiometric calibration of the flight data.

*Response:*

The pre-flight, radiometric calibration of EMI was not conducted under flight-like vacuum and possibly under thermal conditions due to the limitation of the calibration facility. The EMI on-ground response to the quartz tungsten halogen WLS (6 V, 10 W) is displayed in following figure, which uses UV2 and VIS1 as examples.

[Figure]

The EMI in-orbit response to the quartz tungsten halogen will be obtained after the launch. The change between the on-ground and in-orbit responses is used to correct the preflight radiometric calibration, which in turn is used to accomplish the in-flight absolute radiometric calibration of the flight data.

(11) Finally, while the manuscript shows the performance of the instrument on the ground, the reader is not told what the actual performance requirements are. Presumably, the instrument performance requirements are driven by the scientific objectives. Comparing the measured/estimated performance (e.g. SNR) with the mission requirements would make the conclusion much stronger.

*Response:*

We have added the performance requirements to the introduce section and added the on-ground calibration results in the conclusions section.

**Performance requirements**

Spectral range: UV1:240–315 nm; UV2:311–403 nm; VIS1:401–550 nm; VIS2: 545–710 nm;

Spectral resolution: <0.55 nm;

Accuracy of the on-ground wavelength calibration: <0.05 nm;

Accuracy of the on-ground radiometric calibration: <5%;

SNR:

UV channel: >200 (@1.27 $\mu W / cm^2 / sr / nm$)

VIS channel: >1300 (@10.89 $\mu W / cm^2 / sr / nm$)

**Conclusions**

The spectral and radiometric response performance of the EMI is obtained by preflight calibration. The on-ground calibration results are shown as follows:
Spectral calibration results:
UV1: 236.44–317.28 nm with the spectral resolution ≤0.45 nm;
UV2: 306.08–407.12 nm with the spectral resolution ≤0.49 nm;
VIS1: 395.50–552.63 nm with the spectral resolution ≤0.48 nm;
VIS2: 534.63–712.90 nm with the spectral resolution ≤0.49 nm;
The final accuracy of the wavelength calibration is <0.05 nm.
Radiometric calibration results:
UV1: 4.64%, UV2: 4.63%, VIS1: 4.43%, VIS2: 4.42%.

The on-ground calibration results meet the performance requirements of the EMI.

The EMI in-orbit simulation $SNR_{simulation}$ is obtained by the radiance $R_{simulation}$ at an albedo of 0.3 and solar zenith of 60°. The in-orbit simulation SNR at the radiance of 1.27/10.89 $\mu W / cm^2 / sr / nm$ can be achieved by the following equation:

$$SNR = SNR_{simulation} \cdot \sqrt{\frac{R}{R_{simulation}}} ,$$

where $R$ is 1.27 for UV channels and 10.89 $\mu W / cm^2 / sr / nm$ for VIS channels.

For the in-orbit simulation SNR at the radiance of 1.27/10.89 $\mu\mathrm{W}/cm^2/sr/nm$, the results are presented in the following table.

In-orbit simulation SNR at the requirement radiance

| Channel | | SNR (simulation) | SNR (requirements) |
|---------|---------|------------------|---------------------|
| UV2 | 330nm | 328 | 200 |
| | 360nm | 356 | 200 |
| | 390nm | 388 | 200 |
| VIS1 | 420nm | 1860 | 1300 |
| | 480nm | 1900 | 1300 |
| | 540nm | 2040 | 1300 |
| VIS2 | 560nm | 2200 | 1300 |
| | 620nm | 2300 | 1300 |
| | 680nm | 2400 | 1300 |

---

## Referee Comment (RC2) · Anonymous Referee #2 · 29 Jun 2018

I find the response from the authors useful and encourage them to include all the additional information that was provided in the manuscript.

---

## Author Comment (AC2) · 30 Jun 2018

[revised manuscript text omitted]
 possibly because the light source and the dark and readout noises of the pixel vary during the SNR measurement. However, SNR is a good choice for estimating the EMI in-orbit SNR using the SNR model.

The simulation EMI in-orbit SNR of the UV2, VIS1, and VIS2 are displayed in Fig. 15. The in-orbit SNR of this channel is not estimated as the solar light in the band of the UV1 (240 – 310 nm) is absorbed by the atmosphere.

[Figure]

Fig.15. Simulation EMI in-orbit SNR of UV2, VIS1, and VIS2. The input radiance for the SNR model is obtained by MODTRAN with the albedo of 0.3 and with the sun zenith at 60°. The EMI simulation SNR has an integration time of 2 s; it has binning factors of 6 for UV 2 channels and 4 for VIS channels. The spectral bandwidth of a pixel was 0.09 nm for UV2, 0.12 nm for VIS1, and 0.13 nm for VIS2.

The in-orbit radiance obtained by MODTRAN for an albedo of 0.3 at 60° sun zenith is used for the simulation EMI in-orbit SNR. The in-orbit SNR of the UV2 is approximately 700, the VIS1 is approximately 1800, and the VIS2 is about 2000. Under dark scene conditions, the SNR can be improved by increasing 
[revised manuscript text omitted]

---

## Referee Comment (RC3) · Anonymous Referee #1 · 3 Aug 2018

The paper by Zhao et al. reports on the preflight calibration of the Chines Environmental Trace Gases Monitoring Instrument (EMI). Wavelength calibration of the instrument, a thermal vacuum test to investigate the impact of in-orbit conditions on the whole system and the radiometric calibration are described in detail and results are shown. Furthermore, the expected signal-to-noise ratio for each channel has been estimated using model calculations.

This review refers to the modified manuscript submitted by the authors on June 30. The manuscript is in general clearly written and I recommend it for publication in AMT. However, the authors should consider following comments and recommendations.

Section on performance requirements: The authors should give some information on what these requirements based on. I recommend putting the information either in a table or in proper sentences. Please add this section after the general instrument description.

Instrument description: I'm wondering, why the expected spatial resolution in the Visible is smaller than in the UV since the expected intensity should be larger.

Thermal vacuum test: I'm wondering about the relatively small temperature range investigated in this study. Is this really something to expect in reality?

Radiance calibration, Dark signal: The authors stated, that the spectrometer in the Visible has temperature control and changes of the CCD are therefore not an issue. Again, the question: Is this true under real in-orbit conditions e.g. when the system comes from the dark to the illuminated part of the orbit?

SNR (do not use an acronym in the caption): Table 8 and also some sentences concerning the SNR should move from the Conclusions section to the SNR section. In general, I'm a bit unsettled that the assumption of an albedo of 0.3 in the SNR simulations is useful. For most of the relevant scenes the albedo is much lower!

Minor corrections
- Line 11, please change to launch date
- Line 25f: Check sentence for clarity
- Line 29f: Check citations - I recommend to use following publications instead:
  Burrows et al.: The global ozone monitoring experiment (GOME): Mission concept and first scientific results, 1999
  Bovensmann et al.: SCIAMACHY: Mission objectives and measurement modes, 1999
  Levelt et al., The Ozone Monitoring Instrument, 2006
- Line 96: travels instead of travel
- Line 145: …considered as a Gaussian-type function …
- Line 155: … and the accuracy of the FWHM ..
- Line 161f: A mercury argon lamp is used as light source for EMI …
- Figures 5 and 6: What is NTC??
- Line 207: … are presented ...
- Line 224: Write solar calibration mode (SCM) in caption
- Line 279: Table missing?
- Line 308: about 0,5% per what??

- Line 316, Figure 8: … for … instead of … under …
- Line 332: … check sentence for clarity …
- Line 346f: Check numbers given here!!
- Line 358: Based …
- Line 429f: … have been discussed elsewhere …
- Line 450: … are recorded …
- Line 467f: .. of the SNR … and check sentence for clarity
- Line 470f: I'm not sure, what the authors would like to point out here.
- Line 472: The simulation of the … in the UV2, … channels are …
- Line 473: … of channel UV1 …
- L481f: Numbers given here are different to numbers in Table 8!

---

## Author Comment (AC3) · 26 Aug 2018

We would like to thank you for the insightful comments. Our responses to the comments are given below.

General comments:
The paper by Zhao et al. reports on the preflight calibration of the Chines Environmental Trace Gases Monitoring Instrument (EMI). Wavelength calibration of the instrument, a thermal vacuum test to investigate the impact of in-orbit conditions on the whole system and the radiometric calibration are described in detail and results are shown. Furthermore, the expected signal-to-noise ratio for each channel has been estimated using model calculations.

This review refers to the modified manuscript submitted by the authors on June 30. The manuscript is in general clearly written and I recommend it for publication in AMT. However, the authors should consider following comments and recommendations.

(1) Section on performance requirements: The authors should give some information on what these requirements based on. I recommend putting the information either in a table or in proper sentences. Please add this section after the general instrument description.

*Response*:
  Large spectral range from 240 nm to 710 nm combined with high spectral resolution(0.3 nm to 0.5 nm) of the EMI enables the measurement of several trace gases(e.g., NO2, O3, SO2, BrO, HCHO) as well as aerosol, see table 2. To achieve a high retrieval precision, a high SNR is required for the scattered radiance from the UV to the VIS.

Table 2. EMI data products.

| Product Name | Wavelength Band/nm |
| --- | --- |
| O3 | 300-345(UV1,UV2) |
| SO2 | 305-330(UV1,UV2) |
| NO2 | 425-500(VIS1) |
| BrO | 344-360(UV2) |
| HCHO | 335-360(UV2) |
| Aerosol | UV2,VIS1,VIS2 |

(2) Instrument description: I'm wondering, why the expected spatial resolution in the Visible is smaller than in the UV since the expected intensity should be larger.

*Response:*

CCD for the Visible has 576 pixels in the spatial range, each pixel measuring $22.5 \times 22.5\, um^2$. CCD for the UV has 1032 pixels in the spatial range, each pixel measuring $13 \times 13\, um^2$. Calibration results show that:

➢ Spatial resolution in the Visible is 12km on electronic binning of 4, and is 48km on electronic binning of 16.
➢ Spatial resolution in the UV is 8km on electronic binning of 4, and is 48km on electronic binning of 24.

(3) Thermal vacuum test: I'm wondering about the relatively small temperature range investigated in this study. Is this really something to expect in reality?

*Response:*

The in-orbit results showed that temperature stability is better than 0.1K. Actually, the temperature investigated in this study has been applied to EMI after launch.

(4) Radiance calibration, Dark signal: The authors stated, that the spectrometer in the Visible has temperature control and changes of the CCD are therefore not an issue. Again, the question: Is this true under real in-orbit conditions e.g. when the system comes from the dark to the illuminated part of the orbit?

*Response:*

An investigation done after launch shows that the temperature stability is better than 0.1K over one orbit. This temperature variation over the orbit leads to very small change of the background signal.

(5) SNR (do not use an acronym in the caption): Table 8 and also some sentences concerning the SNR should move from the Conclusions section to the SNR section. In general, I'm a bit unsettled that the assumption of an albedo of 0.3 in the SNR simulations is useful. For most of the relevant scenes the albedo is much lower!

*Response:*

a) Table 8 and the sentences concerning the SNR have been moved from the Conclusions section to the SNR section.
b) The SNR at albedo of 0.3 is typical SNR of the EMI. SNR at other albedo can be obtained from the typical SNR by equation(16) in the paper:

$$SNR = SNR_{simulation} \cdot \sqrt{\frac{R}{R_{simulation}}}$$

Minor corrections
• Line 11, please change to launch date
Changed (date:2018.05.09)
• Line 25f: Check sentence for clarity
Modified.
• Line 29f: Check citations - I recommend to use following publications instead:
Burrows et al.: The global ozone monitoring experiment (GOME): Mission concept and first scientific results, 1999
Bovensmann et al.: SCIAMACHY: Mission objectives and measurement modes, 1999
Levelt et al., The Ozone Monitoring Instrument, 2006
Changed
• Line 96: travels instead of travel
Corrected
• Line 145: …considered as a Gaussian-type function …
Corrected
• Line 155: … and the accuracy of the FWHM ..
Corrected
• Line 161f: A mercury argon lamp is used as light source for EMI …
Corrected
• Figures 5 and 6: What is NTC??
Corrected (NTC: No Temperature control)
• Line 207: … are presented ...
Corrected
• Line 224: Write solar calibration mode (SCM) in caption
Corrected
• Line 279: Table missing?
Added
• Line 308: about 0,5% per what??
Updated.
• Line 316, Figure 8: … for … instead of … under …
Corrected
• Line 332: … check sentence for clarity …
Modified.
• Line 346f: Check numbers given here!!
Corrected
• Line 358: Based …
Corrected
• Line 429f: … have been discussed elsewhere …
Corrected
• Line 450: … are recorded …

Corrected

• Line 467f: .. of the SNR … and check sentence for clarity

Modified.

• Line 470f: I'm not sure, what the authors would like to point out here.

Modified

• Line 472: The simulation of the … in the UV2, … channels are …

Corrected

• Line 473: … of channel UV1 …

Corrected

• L481f: Numbers given here are different to numbers in Table 8!

Response: The numbers in L481f are obtained by the radiance at an albedo of 0.3 and solar zenith of 60 °. The numbers in table 8 are obtained by the radiance of 1.27/10.89

$\mu W / cm^2 / sr / nm$